# Co-Ultramicronized Palmitoylethanolamide/Luteolin Restores Oligodendrocyte Homeostasis via Peroxisome Proliferator-Activated Receptor-α in an In Vitro Model of Alzheimer’s Disease

**DOI:** 10.3390/biomedicines10061236

**Published:** 2022-05-26

**Authors:** Roberta Facchinetti, Marta Valenza, Chiara Gomiero, Giulia Federica Mancini, Luca Steardo, Patrizia Campolongo, Caterina Scuderi

**Affiliations:** 1Department of Physiology and Pharmacology “Vittorio Erspamer”, SAPIENZA University of Rome—P. le Aldo Moro, 5, 00185 Rome, Italy; roberta.facchinetti@uniroma1.it (R.F.); marta.valenza@uniroma1.it (M.V.); giuliafederica.mancini@gmail.com (G.F.M.); luca.steardo@uniroma1.it (L.S.); patrizia.campolongo@uniroma1.it (P.C.); 2Epitech Research Group, 35030 Padova, Italy; chiara.gomiero@epitech.it; 3Centro Europeo di Ricerca sul Cervello (CERC), IRCCS Santa Lucia Foundation Rome, 00143 Rome, Italy; 4Università Telematica Giustino Fortunato, 82100 Benevento, Italy

**Keywords:** oligodendrocytes, astrocytes, beta-amyloid, palmitoylethanolamide, luteolin, neuroinflammation, peroxisome proliferator-activated receptors, Alzheimer’s disease

## Abstract

Oligodendrocytes are cells fundamental for brain functions as they form the myelin sheath and feed axons. They perform these critical functions thanks to the cooperation with other glial cells, mainly astrocytes. The astrocyte/oligodendrocyte crosstalk needs numerous mediators and receptors, such as peroxisome proliferator-activated receptors (PPARs). PPAR agonists promote oligodendrocyte precursor cells (OPCs) maturation in myelinating oligodendrocytes. In the Alzheimer’s disease brain, deposition of beta-amyloid (Aβ) has been linked to several alterations, including astrogliosis and changes in OPCs maturation. However, very little is known about the molecular mechanisms. Here, we investigated for the first time the maturation of OPCs co-cultured with astrocytes in an in vitro model of Aβ_1–42_ toxicity. We also tested the potential beneficial effect of the anti-inflammatory and neuroprotective composite palmitoylethanolamide and luteolin (co-ultra PEALut), which is known to engage the isoform alfa of the PPARs. Our results show that Aβ_1–42_ triggers astrocyte reactivity and inflammation and reduces the levels of growth factors important for OPCs maturation. Oligodendrocytes indeed show low cell surface area and few arborizations. Co-ultra PEALut counteracts the Aβ_1–42_-induced inflammation and astrocyte reactivity preserving the morphology of co-cultured oligodendrocytes through a mechanism that in some cases involves PPAR-α. This is the first evidence of the negative effects exerted by Aβ_1–42_ on astrocyte/oligodendrocyte crosstalk and discloses a never-explored co-ultra PEALut ability in restoring oligodendrocyte homeostasis.

## 1. Introduction

Aberrant myelination of neuronal axons is a common hallmark of several neurodegenerative diseases [1]. Physiologically, myelination consists in wrapping axons in a sheath of lipid called myelin, which is able to increase the rate of action potentials along the axon [2]. Thus, any impairment in myelin formation may alter neurotransmission and lead to reduced cognitive functions and dementia [3].

Myelination is performed by oligodendrocytes, brain cells belonging to the macroglia family of glial cells [3]. They originate from oligodendrocyte precursor cells (OPCs), mainly localized in the ventricular zones of the central nervous system (CNS). From there, they migrate to the developing brain, where they become active oligodendrocytes [4]. In the adult brain, the OPCs cell cycle is accelerated in case of myelin loss due to brain injury, aging, or diseases [5], to counteract the damage. 

OPCs survival and proliferation are under the control of another macroglia cell type, namely astrocytes [3,6]. The latter are star-shaped cells able to maintain brain homeostasis at all levels of organization [7,8,9]. Among their innumerable functions, astrocytes contribute to myelination because they support OPCs survival by establishing direct cell–cell contacts as well via secreted mediators [7,10]. Among many, astrocytes release both the fibroblast growth factor (FGF)2 and the transforming growth factor (TGF)-β, two enhancers of OPCs differentiation. During aging or some pathological conditions, astrocytes modify the production of these and other factors, thus hampering important physiological functions, including OPCs differentiation and myelin sheath formation [11,12]. For instance, in multiple sclerosis (MS), astrocytes secrete proinflammatory factors such as the tumor necrosis factor (TNF)-α, interleukin (IL)-1β, and IL-6, negatively affecting OPCs differentiation [10,13,14]. Astrogliosis and demyelination also occur in Alzheimer’s disease (AD) [15,16,17,18]. Brain imaging studies have linked myelin impairments with AD pathogenesis, appointing beta-amyloid (Aβ) cerebral deposition as a possible etiological factor [19,20]. Myelin damage coincides indeed with Aβ accumulation and concurrent impairment in OPCs maturation in the brains of AD patients [21]. To date, the molecular mechanisms involved have not been clarified yet. Very recent in vitro and in vivo evidence reported the effect of Aβ in stimulating the maturation of OPCs and promoting the expression of the myelin basic protein (MBP) [22,23]. In order to contribute to understanding these counterintuitive findings, here we investigated the effects of Aβ_1–42_ on the maturation of OPCs co-cultured with astrocytes, exploring the molecular mechanisms. 

Numerous mediators and receptors and, among these, the peroxisome proliferator-activated receptors (PPARs) seem to play a prominent role in regulating the communication between astrocytes and oligodendrocytes [24,25,26,27]. PPARs represent a family of ligand-activated transcription factors playing a vital role in cellular processes [28]. Three different isoforms have been identified named α, β/δ, and γ. All of them are expressed in oligodendrocytes, astrocytes, and neurons [29]. Molecules acting as agonists of these receptors have demonstrated anti-inflammatory and neuroprotective properties in different in vitro and in vivo models of CNS diseases [30,31,32,33,34,35]. In animal models of demyelinating diseases, such as encephalomyelitis and amyotrophic lateral sclerosis, the administration of PPAR-γ agonists delayed the onset and ameliorated the clinical manifestations of the pathology by mainly exerting anti-inflammatory effects [36]. 

It was recently shown in vitro that a formulation containing palmitoylethanolamide (PEA) co-ultramicronized with the flavonoid luteolin (co-ultra PEALut) promotes OPCs morphological development into mature oligodendrocytes [37,38]. Furthermore, co-ultra PEALut significantly reduced the development of encephalomyelitis in an in vivo experimental model of MS [39]. The co-ultra PEALut mechanism of action was not investigated in either of these studies. We and others have shown that PEA exerts anti-inflammatory and neuroprotective properties through the interaction with the PPAR-α [40,41]. Moreover, our laboratory has demonstrated PEA efficacy in dampening astrocyte reactivity in several preclinical models of AD by engaging the PPAR-α [32,42,43,44]. In the light of this, here we tested the effects of the agonism at PPAR-α by co-ultra PEALut in modulating the proper communication between astrocytes and oligodendrocytes necessary for myelination.

Primary OPCs were cultured on permeable membranes inserted in wells containing primary astrocytes, thus allowing secreted soluble factors to diffuse while preventing physical contact between the two different cell types. Using this model, we tested the possible beneficial effect of co-ultra PEALut treatment in counteracting Aβ_1–42_-induced toxicity. We concurrently verified the involvement of PPAR-α by carrying out experiments in the presence or absence of GW6471, a selective PPAR-α antagonist.

The results of the present study provide the very first evidence indicating that the exposure of primary astrocytes to Aβ_1–42_ impairs some astrocytic functions and changes the differentiation and cellular morphology of co-cultured oligodendrocytes. Moreover, here we show a new and never-explored ability of co-ultra PEALut in counteracting Aβ_1–42_-induced damage on oligodendrocyte homeostasis, exerting some effects through the PPAR-α. Considering that co-ultra PEALut is already approved for human use as a food for special medical purposes, altogether, our findings open new opportunities for the treatment of diseases characterized by myelination impairments.

## 2. Materials and Methods

### 2.1. Primary Cell Cultures

All animal procedures were performed in agreement with the guidelines of the Italian Ministry of Health (D.L. 26/2014) and with the European Parliament directive 2010/63/EU.

Primary astrocytes and OPCs were isolated from cerebral cortices of female Sprague Dawley rat pups (6 pups/experimental group) according to our previous studies [32,40,42] and published protocols [45,46] with minor modifications. Rats at postnatal day 0–2 were sacrificed by decapitation. The cortices were isolated, and the meningeal membranes were removed. Each cortex was mechanically grinded in cold sterile Phosphate Buffer Saline (PBS) 1X added with 10 U/mL penicillin and 100 μg/mL streptomycin and then centrifuged (200× *g* for 30 s). The pellet was resuspended in High-glucose Dulbecco’s modified Eagle’s medium (DMEM) containing papain, L-cysteine, and DNase I. After 60 min (min) in a 37 °C water bath with occasional swirling, an ovomucoid solution (DMEM supplemented with DNase I solution, 1% bovine serum albumin solution (BSA), and trypsin inhibitor) was added for additional 2 min. After centrifugation (400× *g* for 3 min), cells were triturated by pipetting several times in fresh ovomucoid solution. Then, DMEM supplemented with 10% fetal bovine serum (FBS), 100 U/mL penicillin, and 100 μg/mL streptomycin (complete DMEM) was added to the suspension. After centrifugation (400× *g* for 5 min), the obtained mixed glial cells were plated on 75 cm^2^ tissue culture flasks (3,000,000 cells/flask) in complete DMEM and cultured at 37 °C in a humified atmosphere containing 5% CO_2_. Complete DMEM was replaced 24 h and 7 days later. Ten days after isolation, microglial cells were dislodged by shaking flasks for 1 h and discarded. OPCs were separated from astrocytes through an overnight shaking session and plated on semi-porous membranes (0.4 μm pore size; density 100,000 cells/membrane), previously coated with poly-D-lysine hydrobromide, in Sato medium, that is DMEM supplemented with 4% 3,3′,5-triiodo-L-thyronine, 4% L-thyroxine, 200 U/mL penicillin and 200 μg/mL streptomycin, 1% FBS, 2% N2 supplement (Thermo Fisher Scientific, Waltham, MA, USA). Astrocytes were plated (200,000 cells/well) in 24-well plates (Nunc, Thermo Fisher Scientific, Waltham, MA, USA) in 1 mL of complete DMEM. Then, semi-porous membranes containing OPCs were located in the 24-well plates containing astrocytes. After plating, OPCs and astrocytes were allowed to acclimatize and grow for 72 h in the respective media before treatments. Remaining astrocytes were plated at 1,000,000 cells/well in 6-well plates for Western Blot experiments and at 30,000 cells/well on 13 mm glass coverslips located in a 24-well plate for immunofluorescence experiments. The procedure is outlined in Figure 1a.

Otherwise specified, all reagents and plasticware used in the above-described protocol were purchased from Sigma Aldrich, Saint Louis, MO, USA.

### 2.2. Drugs and Schedule of Treatments

Human Aβ_1–42_ was purchased from AnaSpec (Fremont, CA, USA) and dissolved in sterile NaOH 10 mM at a concentration of 221.5 μM following the manufacturer’s instructions. In order to obtain the fibrillary form, Aβ_1–42_ solution was diluted in PBS 1X and let aggregate for 24 h at 37 °C [47,48,49].

The selective PPAR-α antagonist GW6471 was purchased by Tocris Bioscience (Bristol, UK) and dissolved in DMSO, whereas a preparation containing co-ultra PEALut (PEA and Luteolin co-ultramicronized, 10:1 by mass; a kind gift of Epitech Group SpA, Saccolongo, Italy) was dissolved in Pluronic F-68 (Sigma-Aldrich, St. Louis, MO, USA) and sonicated for 20 min in a water bath until complete dissolution.

Primary astrocytes were treated with Aβ_1–42_ 1 μM in the presence or absence of the following substances: co-ultra PEALut (3 µm) and GW6471 (3 μM). Concentrations and schedule of treatments were chosen based on the available literature [23,32,40,50,51] and according to the results of neutral red uptake assays. All reagents were lipopolysaccharide-free.

After 48 h of treatment, astrocytes and oligodendrocytes were collected separately and processed for analyses.

### 2.3. Neutral Red Uptake Assay

Respective astrocyte and oligodendrocyte viability was tested by neutral red uptake assay, as previously described [42]. Cells were bathed for 3 h in 50 μg/mL neutral red solution (Sigma-Aldrich) at +37 °C. Later, cells were rinsed first in Ca^2+^- and Mg^2+^-free PBS and then in a de-staining solution containing ethanol, deionized water, and glacial acetic acid (50:49:1 *v*/*v*). Cell absorbance (540 nm) was read by a microplate spectrophotometer (Epoch, BioTek, Winooski, VT, USA). The proportional number of viable cells was calculated and expressed as percentage to control sample (CTRL).

### 2.4. Protein Extraction and Western Blot Analysis

Western blot analysis was performed on protein lysates of primary astrocytes, following our published protocol [42]. Cells were lysed in ice-cold hypotonic buffer containing 150 mM NaCl, 1 mM EDTA, 50 mM Tris/HCl pH 7.5, 1% triton X-100 complete of inhibitors of protease (0.1 mM leupeptin, 10 μg/mL aprotinin, and 1 mM phenylmethylsulfonyl fluoride, all from Sigma-Aldrich). Homogenates were incubated for 40 min at +4 °C, then centrifuged at 18,440× *g* for 30 min. Supernatants containing proteins were stored at −80 °C until use. An equivalent amount of each sample (50 μg), calculated by Bradford assay, was resolved through 12% acrylamide SDS-PAGE Stain-Free precast gels (Bio-Rad Laboratories, Segrate, Italy). All samples were run concurrently. Proteins were transferred from gels to nitrocellulose papers using a trans-blot SD semidry transfer cell (Bio-Rad Laboratories). After 1 h incubation in a blocking solution containing 5% no-fat dry milk and 0.1% Tween 20 in TBS (TBS-T) at room temperature (Tecnochimica Moderna, Rome, Italy), membranes were left overnight at +4 °C in a blocking solution containing one of the following primary antibodies: rabbit anti-glial fibrillary acidic protein (GFAP) 1:25,000 (Abcam, Cambridge, UK), mouse-anti-glutamine synthetase (GS) 1:500 (Millipore, Burlington, MA, USA), and mouse anti-high mobility group box 1 (HMGB1) 1:1000 (BioLegend, San Diego, CA, USA). The following morning, membranes were rinsed using 0.05% TBS-T and incubated with a blocking solution containing the appropriate secondary horseradish peroxidase (HRP)-conjugated antibody for 1 h at room temperature. Lastly, membranes were washed with T-TBS and briefly bathed in a specific solution of the enhanced chemiluminescence (ECL) kit (GE Healthcare Life Sciences, Milan, Italy). Immunocomplexes were detected by a Chemidoc XRS and analyzed by Image Lab software (Bio-Rad, Hercules, CA, USA). Values were normalized to the total protein content. Experimental conditions are summarized in Table 1.

### 2.5. Real Time-Quantitative Polymerase Chain Reaction (RT-qPCR)

RT-qPCR was carried out as previously described [52] and summed up in Table 2. The TRI-Reagent (Sigma-Aldrich) was added to primary astrocyte samples to extract total mRNA. An equal amount of mRNA (1 μg) from each experimental group was quantified by a D30 BioPhotometer spectrophotometer (Eppendorf AG, Hamburg, Germany) and reverse transcribed using a QuantiTect Reverse Transcription Kit (Qiagen, Valencia, CA, USA). RT-PCR reactions (20 μL) were set mixing specific primers, cDNA (20 ng) with the iTaq Universal SYBR Green Supermix (Bio-Rad, Hercules, CA, USA) and run in 96-well plates using the CFX96 Touch thermocycler (Bio-Rad, Hercules, CA, USA). All samples were run concurrently in triplicate. A two-step thermal protocol was used for 40 cycles (10 s at 95 °C and then 30 s at 60 °C) preceded by a polymerase activation step (3 min at 95 °C). Gene-specific amplification was controlled by including a melting curve analysis of the amplification products at the end of the reaction. The Glyceraldehyde-3-Phosphate Dehydrogenase (GAPDH) was used as reference gene to normalize each target amplicon. Data were analyzed using the Pfaffl method, correcting the delta Ct values for each primer efficiency [53]. All primer sequences are listed in Table 2.

### 2.6. Immunofluorescence

Immunofluorescence was performed according to our previous studies [54]. Briefly, semi-porous membranes containing primary oligodendrocytes were carefully cut out and laid down on glass slides (Thermo Scientific, Rockford, IL, USA). Cells were fixed in 4% paraformaldehyde (Santa Cruz, Dallas, TX, USA) in PBS 1X for 20 min at room temperature. After being washed three times with PBS 1X, they underwent a 10 min bath in PBS 1X with 0.25% Triton X-100 (Sigma-Aldrich) to promote cell permeabilization. Then, cells were submerged for 2 h in a blocking solution containing 5% BSA and 0.25% Triton X-100 in PBS 1X and subsequently incubated overnight at 4 °C with the blocking solution added of the following primary antibodies: rabbit anti-GFAP (1:1500, Abcam), rabbit anti-Olig2 (1:200), and mouse anti-MBP (1:200), both from Santa Cruz. The following day, cells were washed 3 times with PBS 1X and incubated for 1 h with a fresh blocking solution containing the FITC-conjugated goat anti-rabbit IgG (H + L) antibody (1:200) and TRITC-conjugated goat anti-mouse IgG (H + L) antibody (1:200) (Jackson ImmunoResearch, Suffolk, UK). Nuclei were stained with Hoechst (1:500, Thermo Fisher Scientific). Lastly, cells were rinsed 3 times in PBS 1X before being covered with Fluoromount aqueous mounting medium (Sigma-Aldrich) and a glass coverslip. Experimental conditions are summarized in Table 3. The specificity of the signal was confirmed in cells that underwent the same protocol procedure except for the incubation with the primary antibody. 

Data were collected using an Eclipse E600 microscope equipped with Nikon Plan 20× and 40× objectives (Nikon instruments, Rome, Italy) connected to a Qimaging camera (Surrey, BC, Canada), and controlled by the software NIS-Elements-BR 3.2 64 bit (Nikon instruments). Parameters of gain and exposure were kept constant to allow comparisons between samples within the same experiment.

### 2.7. Image Analysis

In order to calculate the percentage of MBP^+^/Olig2^+^ oligodendrocytes, cells were counted manually from six random fields per condition for each independent experiment using the Fiji software by two investigators blinded to treatments. 

The morphology of MBP^+^ oligodendrocytes was analyzed by measuring the surface, expressed in μm^2^, of each individual cell [55].

The complexity of MBP^+^ oligodendrocytes was analyzed by using the Sholl analysis plugin [56,57]. Fluorescent pictures were converted into binary 8-bit images, background subtracted, and the free edges were detected. Starting radius was set at 10 μm and step size at 5 μm, thus creating 5 concentric radii up to 30 µm distance from the soma. The number of cell processes intersecting those circles was plotted as a function of the radial distance from the cell soma. Several descriptors of cell arborization were analyzed, which are the maximum intersections (calculated as the highest number of processes/branches in each cell arbor), the sum of intersections (calculated as the sum of intersections detected at each concentric circle), and the mean of intersections (calculated as the sum of intersections divided by intersecting radii) [23,58,59]. Images were analyzed by an investigator blinded to treatments.

### 2.8. Statistical Analysis

Statistical analysis was performed using GraphPad Prism software version 6.0 (GraphPad Software, San Diego, CA, USA). Data were analyzed by either the Student’s *t*-test, in case just two experimental groups were compared, or one-way analysis of variance (ANOVA) in all other experiments. Upon detection of a main significant effect from ANOVA, a post hoc test was carried out as either the Dunnett’s test, in case groups needed to be compared only to the CTRL, or the Student Newman Keuls (SNK) test in case of multiple comparisons among groups. Differences between mean values were considered statistically significant when *p* < 0.05.

## 3. Results

### 3.1. Characterization of OPCs Maturation In Vitro and Effects of Treatments on the Viability of OPCs and Astrocytes

To monitor the maturation of OPCs in our experimental conditions, we performed a longitudinal immunofluorescence study testing the appearance of the MBP signal. We studied the MBP^+^/Olig2^+^ cell ratio, which is considered an index of the level of maturation of OPCs [60]. Indeed, MBP is the major protein constituent of myelin, and it is produced only by mature oligodendrocytes, whereas Olig2 is a specific transcriptional factor expressed by oligodendrocytes at all developmental stages, from precursors to mature cells [60,61,62]. According to literature data, our results show that primary oligodendrocytes cultured alone survive approximately 10 days and start to express MBP between the second and third day in vitro (DIV). MBP expression progressively increases to reach the highest value at about the seventh DIV (Figure 1b). Provided these data, we decided to study the effects of treatments on the initial phase of OPCs maturation, that is, the third DIV. 

Figure 1c–e shows the results of viability assays performed to choose the appropriate concentrations of treatments to be applied for 48 h to astrocytes. In Figure 1, we also report the effects of the selected concentrations and of the relative vehicles used to solubilize each substance on astrocyte (f) and oligodendrocyte (g) viability. No significant variation *versus* control was observed when co-ultra PEALut or GW6471 were provided alone.

### 3.2. Aβ_1–42_ Triggers Astrocyte Reactivity

To study the effects of human fibrillary Aβ_1–42_ 1 µm on astrocyte function in our transwell culture system, we performed immunofluorescence and western blot experiments analyzing the expression levels of key proteins that are commonly related to a reactive and proinflammatory phenotype [63,64]. Our results show that 48 h treatment with Aβ_1–42_ triggers this astrocytic phenotype as documented by the increased expression of GFAP, GS, and HMGB1 (Figure 2).

### 3.3. Co-Ultra PEALut Prevents the Astrocyte Reactivity and the Reduction in Growth Factors Transcription Induced by Aβ_1–42_ Exposure

Experiments investigated whether co-ultra PEALut exerts beneficial effects in primary astrocytes exposed to Aβ_1–42_, focusing on astrocytic factors able to affect OPCs survival and development. The possible engagement of the PPAR-α in such co-ultra PEALut effects was also explored. 

As shown in Figure 3a–e, the challenge with Aβ_1–42_ triggered both astrocyte reactivity and an inflammatory process, detected by the increase in GFAP, S100B, p50NFκB, IL-6, and IL-1β mRNA compared with control cells. Co-ultra PEALut significantly prevented the Aβ_1–42_-induced raises. Concurrent treatment with the PPAR-α antagonist GW6471 hampered co-ultra PEALut to block Aβ_1–42_-induced transcription of IL-6 and IL-1β, indicating PPAR-α involvement in co-ultraPEALut effects.

Furthermore, our results show that Aβ_1–42_ significantly reduced the gene expression of both astrocytic FGF2 and TGF-β compared with control cells (Figure 3f,g). Co-ultra PEALut treatment significantly prevented such Aβ_1–42_-induced modifications. However, concurrent treatment with GW6471 did not change the observed co-ultra PEALut effect, thus indicating that the PPAR-α was not engaged.3.4. Co-ultra PEALut counteracts the altered MBP^+^/Olig2^+^ ratio triggered by Aβ_1–42_ exposure

Experiments investigated whether exposing primary astrocytes to Aβ_1–42_ affects the maturation of co-cultured OPCs and whether co-ultra PEALut prevents the observed Aβ_1–42_-induced changes. The possible involvement of the PPAR-α in co-ultra PEALut effects was tested too. As shown in Figure 4, we observed an increase in the number of MBP^+^ oligodendrocytes when these cells were co-cultured with Aβ_1–42_-challenged astrocytes but not with the vehicle (Figure 4a,b). Olig2^+^ cell count was not affected by treatment, suggesting that no proliferation or death of oligodendrocytes occurred (Figure 4a,c). Coherently, the MBP^+^/Olig2^+^ ratio increased in oligodendrocytes indirectly exposed to Aβ_1–42_ (Figure 4d). Co-ultra PEALut significantly prevented such an Aβ_1–42_-induced effect (Figure 4a–d). Concurrent treatment with GW6471 did not reverse the co-ultra PEALut effect indicating that other mechanisms of action could be involved.

### 3.4. Co-Ultra PEALut Controls the Oligodendrocyte Morphological Changes Induced by Aβ_1–42_ Exposure

Oligodendrocytes change their cell morphology in various CNS pathological conditions, including AD [65,66,67,68,69,70]. As we observed that Aβ_1–42_ increased MBP^+^ cells, we decided to clarify the Aβ_1–42_ effects by performing a thorough morphological analysis of oligodendrocytes. We analyzed the cell surface area and the number of arborizations of MBP^+^ oligodendrocytes. As shown in Figure 5, we observed a reduction in the cell surface area of oligodendrocytes when these cells were co-cultured with astrocytes challenged with Aβ_1–42_ but not with the vehicle. Co-ultra PEALut significantly prevented such Aβ_1–42_-induced effect. Concurrent treatment with the PPAR-α antagonist GW6471 did not block the effect of co-ultra PEALut, suggesting the involvement of other mechanisms.

The Sholl analysis revealed a significant reduction both in the maximum number of intersections and the sum of intersections of MBP^+^ cell branches when oligodendrocytes were indirectly exposed to Aβ_1–42_ (Figure 6a–d). Similarly, the mean number of intersections was reduced (Figure 6e). Co-ultra PEALut prevented such Aβ_1–42_-induced modifications in oligodendrocyte morphology through PPAR-α involvement. Indeed, GW6471 blocked the observed co-ultra PEALut effect (Figure 6a,c–e). By counting the number of arborizations from the MBP^+^ cell soma outwards every 5 µm (Figure 6f), we found that Aβ_1–42_-exposed oligodendrocytes showed a lower number of arborizations at three distance measures (10, 15, and 20 µm) compared to control cells, reaching the greatest and most significant reduction at 20 µm from the soma (Figure 6f). Co-ultra PEALut treatment was effective in preventing these Aβ_1–42_-induced morphological changes. The co-treatment with GW6471 significantly blunted such co-ultra PEALut effect, indicating an engagement of the PPAR-α in the co-ultra PEALut mechanism of action (Figure 6f).

## 4. Discussion

Neuroglia is an extremely heterogeneous population of cells present in the nervous system. The heterogeneity of glial cells correlates with the multiplicity of functions that they perform. Regardless of their morphology and functions, the common fundamental role of all types of glial cells in the maintenance of CNS homeostasis [71]. Among glial cells, astrocytes have recently gained great attention for their homeostatic support that takes place at all levels of brain organization. Therefore, alterations affecting astrocytes are nowadays considered to be implicated in the etiology or progression of almost all disorders affecting the nervous system, including AD [72,73,74,75]. For this reason, neuropharmacologists look at these cells as possible targets for the development of new therapeutics [76,77]. 

One of the less explored functions of astrocytes is their supportive role in the maturation of oligodendrocytes. Astrocytes establish physical and functional connections with oligodendrocytes, thus allowing, among many other processes, the proper formation of myelin [10]. Changes in this cell–cell communication may alter myelination, impairing neurotransmission, ultimately leading to cognitive decline and dementia [3]. Indeed, aberrant myelination is considered one of the key features of AD [17], although it remains an overlooked field of study. To contribute to filling this gap, here we set up an in vitro model of Aβ_1–42_ toxicity to study the interaction between astrocytes and oligodendrocytes, with particular emphasis on astrocyte reactivity and the release of trophic factors required for OPCs maturation, which is predictive of their ability to form myelin. By using semi-porous inserts to grow the two populations of glial cells separately, we treated astrocytes with human Aβ_1–42_ and studied the resulting changes in the two cell populations, with special reference to OPCs maturation. We observed that 48 h exposure to human fibrillary Aβ_1–42_ induced astrocytes to acquire a reactive and proinflammatory phenotype. We indeed observed an increase in the expression of key proteins related to astrocyte reactivity, such as GFAP, GS, and HMGB1. Similarly, elevated transcriptional levels of p50NFkB, IL-6, and IL-1β, well-known proinflammatory mediators, have been observed. Intriguingly, the elevation of such meditators has already been linked to a toxic environment that damages all cells, including oligodendrocytes [78,79,80]. Furthermore, our data reveal that Aβ_1–42_-exposure causes a significant reduction in the astrocytic transcriptional levels of FGF2 and TGF-β, two factors strictly implicated in OPCs maturation [11,12]. Based on these results and literature evidence indicating Aβ-induced altered myelination [81,82], we predicted to observe a defective maturation of co-cultured OPCs in our experimental condition. Our immunofluorescence experiments instead revealed a higher number of MBP^+^ cells and MBP^+^/Olig2^+^ ratio in oligodendrocytes indirectly exposed to human Aβ_1–42_ compared with vehicle. In agreement with these counterintuitive results, direct treatment of oligodendrocytes with oligomeric Aβ_1–42_ promoted MBP upregulation in primary rat oligodendrocyte cultures [23]. Furthermore, a marked increase in oligodendrocyte differentiation was found in the corpus callosum and hippocampus of transgenic models of AD, namely 3xTg-AD and APP/PS1 mice, compared with the wild-type counterparts [83]. As suggested by some researchers, a possible explanation of the Aβ effect on oligodendrocyte MBP expression could be linked to the reaction of these cells to the toxic stimulus [5]. Oligodendrocytes, indeed, may react by increasing their rate of differentiation in the attempt to restore the lost homeostasis [84,85]. By performing an in-depth morphological analysis, here we unveil that this push to the maturation operated by the toxic insult with Aβ_1–42_ actually causes severe morphological changes to oligodendrocytes. We observed that MBP^+^ oligodendrocytes lose their proper dimension and complexity as a consequence of the toxic insult. A significant reduction in the cell surface area, the maximum number of intersections, and both the sum and mean of intersections of MBP^+^ cell branches were detected in oligodendrocytes co-cultured with astrocytes challenged with Aβ_1–42_. All these structural changes highlight oligodendrocyte loss of planarity and their confused organization of branches. Data obtained agree with some recent in vivo findings in 3xTg-AD mice. In particular, oligodendrocytes of six-month-old transgenic mice appeared atrophic when compared to non-transgenic age-matched animals [55].

The new and extremely interesting evidence that emerges from the present results concerns the ability of co-ultra PEALut to restore oligodendrocyte homeostasis impaired by the Aβ_1–42_ challenge. Our and other groups have extensively documented the neuroprotective and beneficial effects of PEA in several preclinical and clinical settings [41,86,87,88]. We also demonstrated the ability of both PEA and co-ultra PEALut to modulate glial reactivity and dampen neuroinflammation, thus fostering neuron survival in different AD models [32,40,42,43,44]. 

Here, we provide the very first evidence about co-ultra PEALut’s ability to restore the pathological changes occurring in both astrocytes and oligodendrocytes after a toxic insult with the human Aβ_1–42_. We also show that co-ultra PEALut exerts some of these effects with a mechanism mediated by the PPAR-α. The activation of PPAR-α has been linked to NFκB inhibition, reduction of proinflammatory mediators, increased transcription of antioxidant enzymes, and promotion of cell survival through the inhibition of caspase 3 [89]. All these pathways could theoretically explain our results; however, further experiments are needed to clarify the effects of co-ultra PEALut. 

By increasing FGF2 and TGF-β mRNA levels and blunting the expression of proinflammatory mediators, co-ultra PEALut normalized astrocyte functions and preserved the morphological features of oligodendrocytes indirectly exposed to Aβ_1–42_. It is worth noting that co-ultra PEALut provided concurrently with Aβ_1–42_, apart from preventing Aβ_1–42_-induced branching loss throughout the entire cell, even increases the number of intersections at 20 µm from the soma compared with the control group. Interestingly, the total area occupied by oligodendrocytes and the number of cellular arborizations are associated with the ability of these cells to create the myelin sheath around neuronal axons [90,91]. We are aware that further studies are required to better clarify the molecular mechanisms. However, our data are promising as they suggest the potential use of co-ultra PEALut in neurological disorders characterized by impaired myelination. In agreement, recent studies showed the co-ultra PEALut’s ability to promote in vitro OPCs differentiation [37,38,46] and to reduce the development of clinical signs in an in vivo experimental model of MS [39]. 

This study has some limits: first, our in vitro system is not able to fully reproduce the physiological environment where myelination occurs, which also comprehends neurons. Secondly, the mechanism of action of co-ultra PEALut could be further explored since not all the observed effects here are mediated by the PPAR-α. Indeed, other receptors have been proposed as PEA targets, including the transient receptor potential vanilloid type 1 channel and the orphan G-protein coupled receptor 55; PEA could engage the endocannabinoid enzyme fatty acid amide hydrolase acting as a false substrate and, in turn, indirectly increasing anandamide concentration [77,92]. Finally, further studies are needed to characterize additional markers targeting myelin and to explore OPCs differentiation and maturation longitudinally.

In conclusion, our data show a novel and never-explored ability of co-ultra PEALut in counteracting the negative effects exerted by Aβ on oligodendrocyte homeostasis. Altogether, our findings open new opportunities for the adjuvant treatment of AD, and co-ultra PEALut is a promising composite capable of exerting beneficial effects in both experimental and clinical settings. Since it is already licensed for human use, the potential to proceed to clinical use rapidly is reasonable.

## Figures and Tables

**Figure 1 biomedicines-10-01236-f001:**
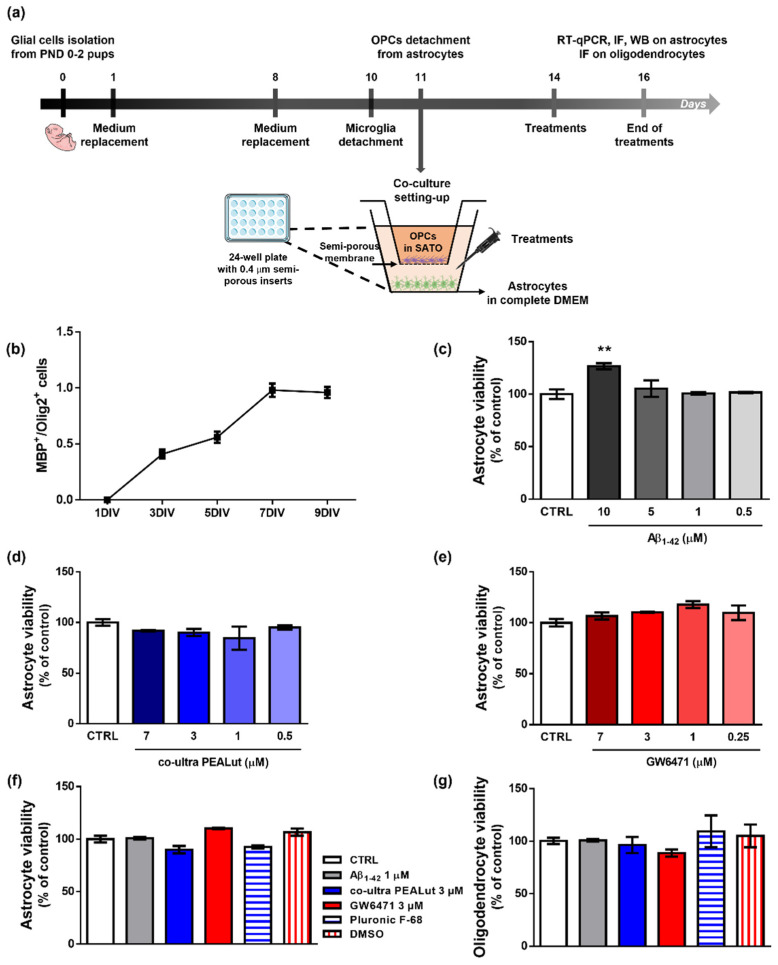
Experimental design, characterization of OPCs maturation and effects of treatments. (**a**) Schematic representation of the experimental design. (**b**) MBP^+^/Olig2^+^ cell count at 1, 3, 5, 7, and 9 days in vitro (DIV). (**c**) Astrocyte viability after 48 h treatment with human fibrillary Aβ_1–42_ (0–0.5–1–5–10 µm). (**d**) Astrocyte viability tested by neutral red uptake assay after 48 h treatment with co-ultra PEALut (0–0.5–1–3–7 µm). (**e**) Astrocyte viability tested by neutral red uptake assay after 48 h treatment with GW6471 (0–0.25–1–3–7 µm). (**f**) Astrocyte and (**g**) oligodendrocytes viability after 48 h treatment with either complete DMEM or Sato medium (CTRL), Aβ_1–42_ 1 µm, co-ultra PEALut 3 µm, GW6471 3 µm, Pluronic F-68, and DMSO. Data are plotted as mean ± SEM and expressed as percentage variations to the control group (CTRL). ** *p* < 0.01 vs. CTRL (Dunnett’s test).

**Figure 2 biomedicines-10-01236-f002:**
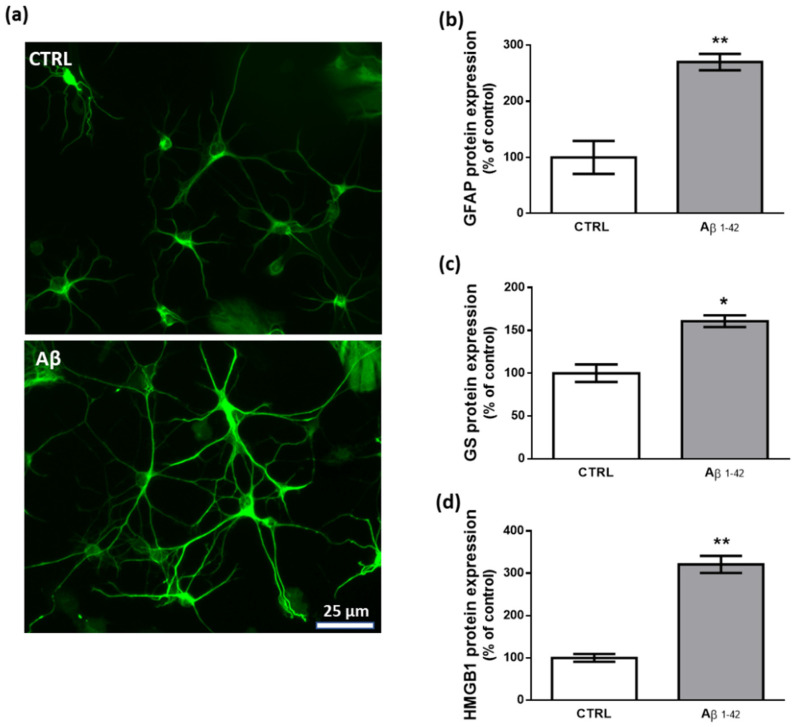
Effects of Aβ_1–42_ on astrocytes. (**a**) Representative photomicrographs of GFAP immunolabeled cells. Results from western blot experiments show increased expression of GFAP (**b**), GS (**c**), and HMGB1 (**d**) in astrocytes treated with human fibrillary Aβ_1–42_ 1 µm for 48 h compared with vehicle-treated cells (CTRL). Data are plotted as mean ± SEM and expressed as percentage variations vs. CTRL. * *p* < 0.05; ** *p* < 0.01 vs. CTRL (Student’s *t*-test).

**Figure 3 biomedicines-10-01236-f003:**
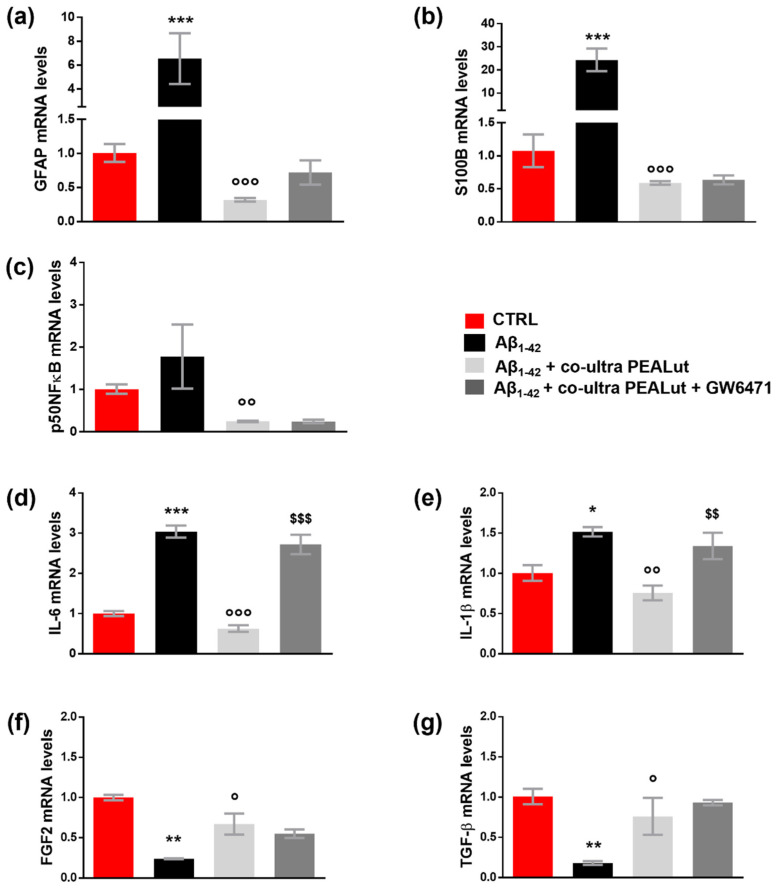
Co-ultra PEALut blunts the Aβ_1–42_-induced reactive and proinflammatory astrocyte phenotype. Astrocytic transcriptional changes of GFAP (**a**) and S100B (**b**), two markers of astrocyte reactivity, proinflammatory mediators p50NFκB (**c**), IL-6 (**d**), and IL-1β (**e**), and growth factors FGF2 (**f**), and TGF-β (**g**). Treatments with human fibrillary Aβ_1–42_ 1 µm in the presence or absence of co-ultra PEALut 3 µm and GW6471 3 µm (or respective vehicles) were added to the culture media of primary astrocytes set in co-cultures with OPCs. Primary astrocytes were collected 48 h later and processed for RT-qPCR. Data are expressed as mean ± SEM of the fold change to the vehicle-treated group (CTRL). * *p* < 0.05; ** *p* < 0.01; *** *p* < 0.001 vs. CTRL; ° *p* < 0.05; °° *p* < 0.01; °°° *p* < 0.001 vs. Aβ_1–42_; $$ *p* < 0.01; $$$ *p* < 0.001 vs. Aβ_1–42_ + co-ultra PEALut (SNK test).

**Figure 4 biomedicines-10-01236-f004:**
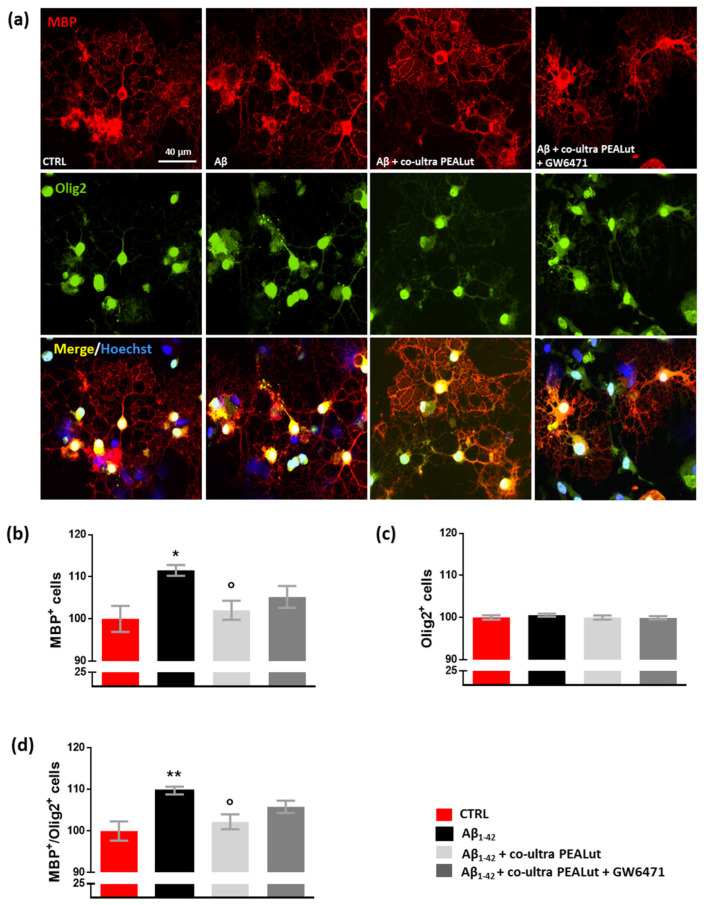
Co-ultra PEALut dampens the altered MBP^+^/Olig2^+^ ratio caused by Aβ_1–42_. (**a**) Representative photomicrographs of MBP (red) and Olig2 (green) staining of primary oligodendrocytes obtained using a 20× objective mounted on a fluorescent microscope. Cell nuclei were stained with Hoechst (blue). Scale bar is 40 μm. (**b**) MBP^+^ cell count. (**c**) Olig2^+^ cell count. (**d**) The ratio of MBP^+^ to Olig2^+^ cell counts. Treatments with human fibrillary Aβ_1–42_ 1 µm in the presence or absence of co-ultra PEALut 3 µm, GW6471 3 µm (or respective vehicles) were added to the astrocytic culture medium. Semi-permeable inserts containing the primary oligodendrocytes were collected 48 h later and processed by immunofluorescence. Data are plotted as mean ± SEM and expressed as percentage variations vs. the vehicle-treated group (CTRL). * *p* < 0.05; ** *p* < 0.01 vs. CTRL; ° *p* < 0.05 vs. Aβ (SNK test).

**Figure 5 biomedicines-10-01236-f005:**
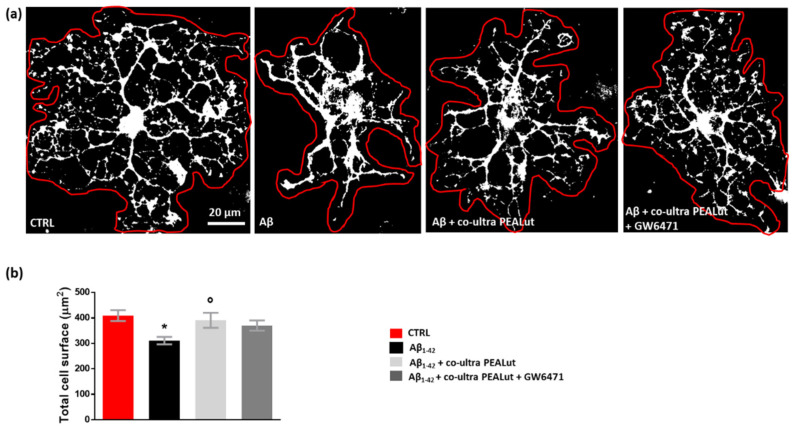
Co-ultra PEALut prevents oligodendrocyte shrinkage caused by Aβ_1–42_. (**a**) Representative thresholded (white color) photomicrographs of MBP^+^ cells taken under a 40× objective mounted on a fluorescent microscope. Scale bar is 20 μm. (**b**) Quantification of total MBP^+^ cell surface area expressed in µm^2^. Cell contours were manually drawn using Fiji. Data are plotted as mean ± SEM. * *p* < 0.05 vs. CTRL; ° *p* < 0.05 vs. Aβ (SNK test).

**Figure 6 biomedicines-10-01236-f006:**
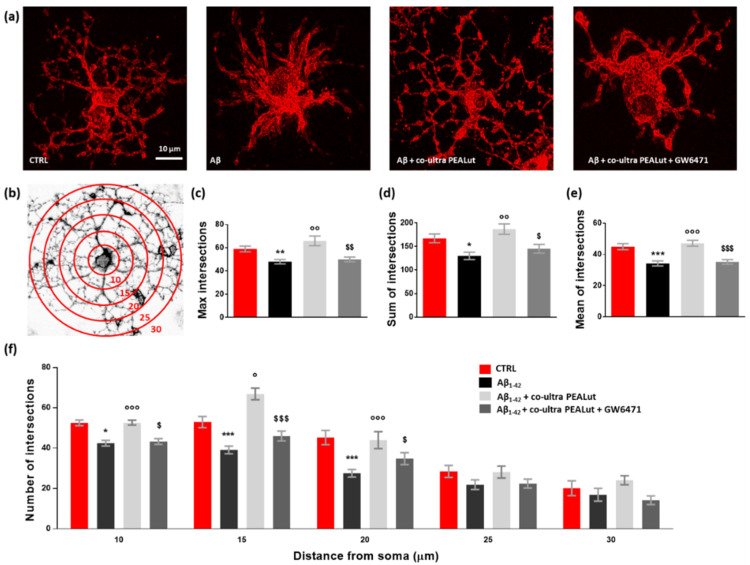
Co-ultra PEALut limits the Aβ_1–42_-induced alterations in oligodendrocyte morphology. (**a**) Representative photomicrographs of MBP^+^ cells taken using a 40× objective mounted on a fluorescent microscope. Scale bar is 10 μm. (**b**) Eight-bit image of a representative MBP^+^ cell, used in the Sholl analysis, in which concentric circles were juxtaposed on each MBP^+^ cell starting from 10 μm far from the cell soma and spaced 5 μm apart. (**c**) Maximum number of intersections, that is the highest number of processes/branches counted in each MBP^+^ cell. (**d**) Sum of intersections, that is the number of intersections counted in each circle. (**e**) Mean of intersections, that is the ratio sum of intersections /number of circles. (**f**) Number of intersections counted every 5 µm starting from 10 μm far from the cell soma until 30 μm. At least 20 cells per animal were counted. Data are plotted as mean ± SEM. * *p* < 0.05; ** *p* < 0.01; *** *p* < 0.001 vs. CTRL; ° *p* < 0.05; °° *p* < 0.01; °°° *p* < 0.001 vs. Aβ_1–42_; $ *p* < 0.05; $$ *p* < 0.01; $$$ *p* < 0.001 vs. Aβ_1–42_ + co-ultra PEALut (SNK).

**Table 1 biomedicines-10-01236-t001:** Western blot conditions.

Primary Antibody	Brand	Dilution	Secondary Antibody	Brand	Dilution
Rabbit α-GFAP	Abcam	1:25,000 5% milk in TBS-T 0.1%	HRP conjugated goat anti-rabbit IgG	Jackson ImmunoResearch	1:10,000 5% milk in TBS-T 0.1%
Rabbit α-GS	Millipore	1:500 5% milk in TBS-T 0.1%	HRP conjugated goat anti-mouse IgG	Jackson ImmunoResearch	1:10,000 5% milk in TBS-T 0.1%
Rabbit α-HMGB1	BioLegend	1:1000 5% milk in TBS-T 0.1%	HRP conjugated goat anti-mouse IgG	Jackson ImmunoResearch	1:10,000 5% milk in TBS-T 0.1%

GFAP: glial fibrillary acidic protein; GS: glutamine synthetase; HMGB1: high mobility group box 1; TBS-T: tris buffered saline tween 20; HRP: horseradish peroxidase.

**Table 2 biomedicines-10-01236-t002:** RT-qPCR parameters and list of primers.

Gene	Brand	Primer (5′ → 3′)		Ann. (60 °C)	Efficiency (%)	R^2^
IL-1β	Bio-Rad	Forward	N/A (Cod. qRnoCID0004680)	60	98.0	0.999
		Reverse				
IL-6	Sigma Aldrich	Forward	CAGAGTCATTCAGAGCAATAC	60	96.1	0.997
		Reverse	CTTTCAAGATGAGTTGGATGG			
TGF-β	Bio-Rad	Forward	N/A (Cod. qRnoCID0006448)	60	102.0	0.999
		Reverse				
FGF2	Bio-Rad	Forward	N/A (Cod. qRnoCID0003540)	60	96.0	0.999
		Reverse				
p50NFκB	Bio-Rad	Forward	N/A (Cod. qRnoCID0003698)	60	92.0	0.997
		Reverse				
GFAP	Sigma Aldrich	Forward	CGGCTCTGAGAGAGATTCGC	60	101.7	0.994
		Reverse	GCAAACTTGGACCGATACCA			
S100B	Bio-Fab Research (Roma, Italy)	Forward	TCAGGGAGAGAGGGTGACAA	60	106.4	0.994
		Reverse	ACACTCCCCATCCCCATCTT			
GAPDH	Bio-Fab research	Forward	GCGAGATCCCGCTAACATCAAAT	60	100.0	0.999
		Reverse	GCCATCCACAGTCTTCTGAGTGG			

Ann.: annealing; IL: interleukin; TGF-β: transforming growth factor-β; FGF2: fibroblast growth factor 2; NFκB: nuclear factor kappa-light-chain-enhancer of activated B cells; GFAP: glial fibrillary acidic protein; GAPDH: glyceraldehyde-3-phosphate dehydrogenase.

**Table 3 biomedicines-10-01236-t003:** Immunofluorescence conditions.

Primary Antibody	Brand	Dilution	Secondary Antibody	Brand	Dilution
Rabbit α-Olig2	Santa Cruz	1:200 5% BSA in PBS/0.25% Triton X-100	FITC conjugated goat anti-rabbit IgG (H + L)	Jackson ImmunoResearch	1:200, 5% BSA in PBS/0.25% triton X-100
Mouse α-MBP	Santa Cruz	1:200 5% BSA in PBS/0.25% Triton X-100	TRITC conjugated goat anti-mouse IgG (H + L)	Jackson ImmunoResearch	1:200, 5% BSA in PBS/0.25% triton X-100
Mouse α-GFAP	Abcam	1:1500 5% BSA in PBS/0.25% Triton X-100	FITC conjugated goat anti-rabbit IgG (H + L)	Jackson ImmunoResearch	1:200, 5% BSA in PBS/0.25% triton X-100

MBP: myelin basic protein; GFAP: glial fibrillary acidic protein; BSA: bovine serum albumin; FITC: fluorescein isothiocyanate; TRITC: tetramethyl rhodamine; PBS: phosphate buffer saline.

## Data Availability

The data that support the findings of this study are available from the corresponding author upon reasonable request.

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
