# Peer review of "Co-Ultramicronized Palmitoylethanolamide/Luteolin Restores Oligodendrocyte Homeostasis via Peroxisome Proliferator-Activated Receptor-α in an In Vitro Model of Alzheimer’s Disease"

_biomedicines, 2022, doi:10.3390/biomedicines10061236_

Round 1
Reviewer 1 Report
The paper presents an experimental study of the maturation of OPCs co-cultured with astrocytes in an in vitro model of Aβ1-42 toxicity. Effects of co-ultra 25 PEALut are also investigated. The results show the negative impact of Aβ1-42 on astrocyte/oligodendrocyte crosstalk and the fact that co-ultra PEALut partially counteracts the Aβ1-42-induced effects. The experiments and analysis are well done and the paper is very well written. The representative photomicrographs shown in some figures look impressive. The few minor issues that the authors should address in the paper in order to improve the overall presentation are as follows.
1. In subsection 2.1, it says that 'Sprague Dawley 110 rat pups (6 pups/experimental group)' were used. Please clarify the following:
1.1 How many experimental groups were used?
1.2 What sex where the pups?
1.3 Are pups appropriate for studying Alzheimer disease for which ageing is a primary risk factor?
2. In subsection 2.7 it says that 'cells were counted manually from six random fields per condition for each independent experiment using the Fiji software.' How was a possible error caused by a manual count dealt with? Did the same person do all the counting?
3. In subsection 2.8 it says that 'data were analyzed by either Student’s t test or one-way analysis of variance (ANOVA) as appropriate.', and 'multiple comparisons were carried out by either the Student Newman Keuls (SNK) or Dunnett’s post-hoc test.' Please briefly explain when the ANOVA test was used instead of the t-test and why. Similarly, please explain when the SNK test was used instead of the post-hoc test and why.
4. How were the red contours in Fig. 5(a) obtained? Presumably, these contours are used to calculate total surfaces of cells.
5. Was the Fiji software used to perform the Sholl analysis? Do any of figs.6(c)-(f) show branching indexes (Garcia-Segura, LM, Perez-Marquez, J, J. Neurosci. Methods 226: 103-109, 2014)?
Author Response
Our replies
We thank Reviewer#1 for the positive comments on our study. Please find below the answers point-by-point.
- In subsection 2.1, it says that 'Sprague Dawley rat pups (6 pups/experimental group)' were used. Please clarify the following:
1.1 How many experimental groups were used?
We sacrificed 6 female pups and obtained 6 independent cultures that we cultured in different supports (i.e., 96-24- and 6-well plates). Using these cells, we set all the experiments.
The number of experimental groups varied depending on the assay.
For instance, Neutral Red Uptake assays shown in Figure 1c-e include five experimental groups since we tested the effects of different concentrations of a single molecule (4 concentrations versus control group) or, as shown in Figure 1f-g, six experimental groups since we tested the effects of the chosen concentrations of each molecule and the relative vehicles respect to control group on cell viability.
1.2 What sex were the pups?
We decided to use female rats following the latest EMA recommendation for in vitro testing and because females are more likely to develop Alzheimer’s disease. We have now included this information in the text.
1.3 Are pups appropriate for studying Alzheimer disease for which ageing is a primary risk factor?
We thank Reviewer#1 for raising this interesting point. Rodent primary brain cells challenged with Aβ are considered a valid model to study Alzheimer’s disease in vitro. Unfortunately, primary neuroglia cultures give a low yield when isolated from adult animals. Cells are usually isolated from embryos or neonates to maximize the yield.
- In subsection 2.7 it says that 'cells were counted manually from six random fields per condition for each independent experiment using the Fiji software.' How was a possible error caused by a manual count dealt with? Did the same person do all the counting?
To manage possible errors in cell counting, experiments were carried out by two researchers blinded to treatments. We thank Reviewer#1 for the comment, we have now included this information in the paragraph Image analysis.
- In subsection 2.8 it says that 'data were analyzed by either Student’s t test or one-way analysis of variance (ANOVA) as appropriate.', and 'multiple comparisons were carried out by either the Student Newman Keuls (SNK) or Dunnett’s post-hoc test.' Please briefly explain when the ANOVA test was used instead of the t-test and why. Similarly, please explain when the SNK test was used instead of the post-hoc test and why.
We apologize for the lack of clarity; we re-written part of paragraph 2.8 “Statistical analysis” of the method section as follows:
“Data were analyzed by either the Student’s t test, in case just two experimental groups were compared, or one-way analysis of variance (ANOVA) in all other experiments. Upon detection of a main significant effect from ANOVA, a post hoc test was carried out as either the Dunnett’s test, in case groups needed to be compared only to CTRL, or the Student Newman Keuls (SNK) test in case of multiple comparisons among groups. Differences between mean values were considered statistically significant when p < 0.05”.
- How were the red contours in Fig. 5(a) obtained? Presumably, these contours are used to calculate total surfaces of cells.
We used the Fiji software tool to manually draw cell contours. This information is now included in the corresponding legend to the figure.
5. Was the Fiji software used to perform the Sholl analysis? Do any of figs.6(c)-(f) show branching indexes (Garcia-Segura, LM, Perez-Marquez, J, J. Neurosci. Methods 226: 103-109, 2014)?
We used the Fiji software tool to perform the Sholl analysis. The J. Neurosci. Method paper cited by Reviewer#1 relates to branching indexes studied in neurons. We are not aware that this protocol could be applied to oligodendrocyte morphology. Although it would be fascinating to try it, this is something that would need extensive validation that is out of the scope of the present paper. We instead followed the method published in Ferreira et al. Nat Methods, 2014. 11(10): p. 982-4 and applied to oligodendrocytes by Quintela Lopez et al. Cell Death Dis, 2019. 10(6): p. 445.
Reviewer 2 Report
The paper entitled "Co-ultramicronized Palmitoylethanolamide/Luteolin restores 2 oligodendrocyte homeostasis via peroxisome proliferator-acti-3 vated receptor-α in an in vitro model of Alzheimer’s disease" by Roberta Facchinetti1 et al presents a very interesting work. In a co-culture model of oligodendrocytes and astrocytes cultured in the presence of amyloid beta 1-42 associated or not with the composite co-ultra-PEALut, the cytoprotective activity of co-ultraPEALut on Abeta 1-42 induced oligodendrocytes and astrocytes changes are clearly shown.
1) Co-ultraPEALut blunts the abeta1-42 induced reactive and pro-inflammatory astrocyte phenotype
2) Co-ultraPEALuts dampens the altered MBP / Oligo2 ratio caused by Abeta1-42
3) Co-ultra PEALut prevents oligodendrocytes shrinkage caused by Abeta1-42
4) Co-ultra PEALut limits the Abeta1-42 induced alterations in oligodendrocyte morphology
The different parts of the manuscript are well presented and the illustrations are of high qualities.
However, the ability of Co-ultra PEALut to limit the Abeta1-42 induced oligodendrocyte alterations for the different parameters studied is not always clear depending on the parameter considered. This must be mentioned in the abstract, in the results and more deeply discussed in the discussion.
In addition, with the use of the PPARalpha antagonist GW6471, it is not always possible to affirm that PPARalpha dependent mecanisms are involved. This must be mentioned in the results. This must be also more deeply discussed in the discussion.
Author Response
We thank Reviewer#2 for the positive comments on our study and the suggestions that we believe have contributed to improving our manuscript.
Our replies point-by-point
1. However, the ability of Co-ultra PEALut to limit the Abeta1-42 induced oligodendrocyte alterations for the different parameters studied is not always clear depending on the parameter considered. This must be mentioned in the abstract, in the results and more deeply discussed in the discussion.
We agree that PEALut strongly controls inflammation and astrocyte reactivity while its effect on oligodendrocyte morphology is milder. Following Reviewer#2 suggestions, we have now added new sentences into the abstract, results, and discussion sections.
In addition, with the use of the PPARalpha antagonist GW6471, it is not always possible to affirm that PPARalpha dependent mecanisms are involved. This must be mentioned in the results. This must be also more deeply discussed in the discussion.
We agree with Reviewer#2 that the presence (or lack) of PPARα involvement in co-ultraPEALut effect must be specified in the results, and we added some sentences where needed. Now in all sub-paragraphs of the Results section there is mention of this.
Also, according to Reviewer#2 critiques, we discussed in the discussion the possible receptors that could mediate co-ultraPEALut effects besides the PPARα, and also the possible mechanism through which the interaction with the PPARα could mediate PEA’s effect.